

# A histological analysis of coloration in the Peruvian mimic poison frog (*Ranitomeya imitator*)

Mallory de Araujo Miles[1], Mikayla Joyce Johnson[1], Adam M. M. Stuckert[2] and Kyle Summers[1]

[1] Biology Department, East Carolina University, Greenville, NC, United States
[2] Department of Biology and Biochemistry, University of Houston, Houston, TX, United States

Corresponding author
Kyle Summers, summersk@ecu.edu

## ABSTRACT

Aposematism continues to be a phenomenon of central interest in evolutionary biology. The life history of the mimic poison frog, *Ranitomeya imitator*, relies heavily on aposematism. In order for aposematic signals to be effective, predators must be able to learn to avoid the associated phenotype. However, in *R. imitator*, aposematism is associated with four different color phenotypes that mimic a complex of congeneric species occurring across the mimic frog's geographic range. Investigations of the underlying mechanics of color production in these frogs can provide insights into how and why these different morphs evolved. We used histological samples to examine divergence in the color production mechanisms used by *R. imitator* to produce effective aposematic signals across its geographic range. We measured the coverage of melanophores and xanthophores (the area covered by chromatophores divided by total area of the skin section) in each color morph. We find that morphs that produce orange skin exhibit a higher coverage of xanthophores and lower coverage of melanophores than those that produce yellow skin. In turn, morphs that produce yellow skin exhibit a higher coverage of xanthophores and lower coverage of melanophores than those that produce green skin. Generally, across the morphs, a high ratio of xanthophores to melanophores is associated with colors of brighter spectral reflectance. Together, our results contribute to the understanding of color production in amphibians and document divergence in the histology of a species that is subject to divergent selection associated with aposematism.

## INTRODUCTION

The phenomenon of aposematism (the use of conspicuous coloration by prey items to signal toxicity to predators) has long held the interest of ecologists and evolutionary biologists (*Poulton, 1898*; *Ruxton et al., 2019*). Because aposematic signals act as a defense that directly affects predation rate, aposematic organisms are under significant evolutionary pressure to develop color production mechanisms suitable to the predators in their environment (*Seymoure et al., 2018*; *Yeager et al., 2012*).

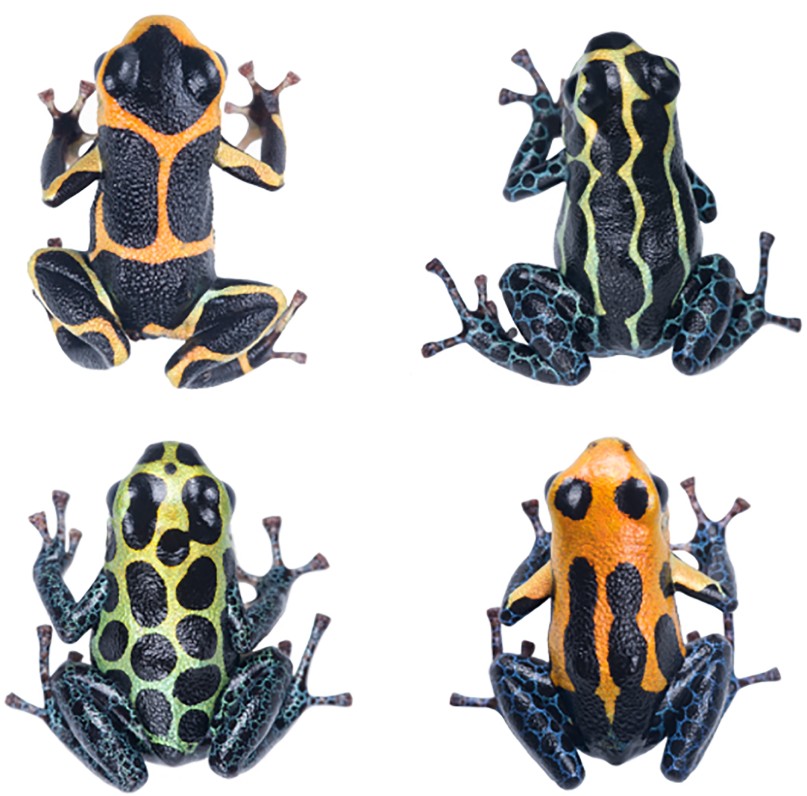

**Figure 1** **Representative color morphs of the mimic poison frog.** The four color morphs of *R. imitator*: banded (upper left), striped (upper right), spotted (lower left), and Varadero (lower right).

The mimic poison frog, *Ranitomeya imitator*, is a small dendrobatid frog endemic to Peru (*Schulte, 1986*). Like many poison frogs, *R. imitator* uses bright aposematic colors to signal its toxicity to potential predators (*Stuckert et al., 2014*; *Stuckert, Venegas & Summers, 2014*; *Summers & Clough, 2001*). Additionally, *R. imitator* provides a striking example of color polytypism. Rather than converging on a single species-specific color pattern, *R. imitator* has diverged into four distinct color morphs—banded, striped, spotted, and Varadero—across its geographic range (Fig. 1).

The evolution and persistence of the four color morphs can be explained by Müllerian mimicry (*Symula, Schulte & Summers, 2001*; *Twomey et al., 2013*; *Chouteau & Angers, 2012*). Each morph benefits from taking on a different color pattern because that color pattern resembles another toxic congener—respectively, *R. summersi*, *R. variabilis* highland, *R. variabilis* lowland, and *R. fantastica*—that shares the same geographic space. The shared color pattern contributes to reciprocal learned avoidance in local predators (*Stuckert et al., 2014*); a predator who has been exposed to a toxic model will, in the future, avoid preying upon the mimic and vice versa.

## Color production

Most color production mechanisms in vertebrates can be categorized either as structural or pigmentary. Structural mechanisms produce bright colors, often blues and greens, by

reflecting light off nanoscale structures found in the integument. By contrast, pigments produced by specialized cells in the dermis absorb light of a specific wavelength, leaving the remaining wavelengths visible to an observer (*Mills & Patterson, 2008*). Both mechanisms frequently interact to produce colors of varying hue and brilliance (*Segami Marzal et al., 2017*).

Amphibians use specialized cells called chromatophores to produce color. Chromatophores are found layered between the epidermal and dermal tissue (Fig. 2). Chromatophore layers are usually found in close contact with each other and are frequently referred to as chromatophore units (*DuShane, 1935*). The most superficial chromatophore, the xanthophore, contains pteridine and/or carotenoid pigments. These pigments absorb violet, blue, and green light to produce yellow, orange, and red coloration (*Bagnara, Taylor & Hadley, 1968*; *Frost & Robinson, 1984*; *Twomey et al., 2020*). Iridophores may be found below xanthophores. Iridophores contain nanoscale guanine platelets that reflect light; traditionally, they have been associated with the production of bright blue and green coloration, although more recently they have been associated with a broader color spectrum (*Bagnara, Taylor & Hadley, 1968*; *Frost & Robinson, 1984*; *Twomey et al., 2020*). When found beneath xanthophores, iridophores may also increase the brightness of superficial colors (*Shawkey & Hill, 2005*; *Shawkey & d'Alba, 2017*; *Twomey et al., 2020*). The deepest chromatophore, the melanophore, contains eumelanin or pheomelanin pigments. These pigments absorb most of the light in the visible spectrum and produce dark brown or black coloration (*Bagnara, Taylor & Hadley, 1968*; *DuShane, 1935*; *Frost & Robinson, 1984*). When found below iridophores, melanophores may enhance blue or green colors by absorbing stray light scattered incoherently by the guanine platelets (*Shawkey & Hill, 2005*; *Shawkey & d'Alba, 2017*).

Previous studies have associated color pattern variation in the four morphs of *R. imitator* with variation in the types of pigments stored in chromatophores and the arrangement of guanine platelets in chromatophores (*Twomey et al., 2020*). Meanwhile, studies of other anuran species have associated color pattern variation with the relative coverage of melanophores, xanthophores, and iridophores rather than with the pigments and subcellular structures found within chromatophores (*Frost & Robinson, 1984*; *Posso-Terranova & Andres, 2017*). To date, the relative coverage of chromatophore type across the four color morphs of *R. imitator* has not been compared, leaving a gap in the understanding of color production in *R. imitator*.

## Study objectives

Our study aims to determine whether chromatophore coverage varies across the four color morphs of *R. imitator*. We used histological samples to measure the coverage of melanophores and xanthophores in the morphs of *R. imitator* and compared the coverage of different chromatophores across morphs. Melanophore coverage was not expected to vary in black skin sections across color morphs, since all four morphs use a similar black background color to display their brighter patterns. However, xanthophore and melanophore coverage were expected to vary in colored skin sections across color morphs, since each morph displays a different color pattern. Understanding variation in

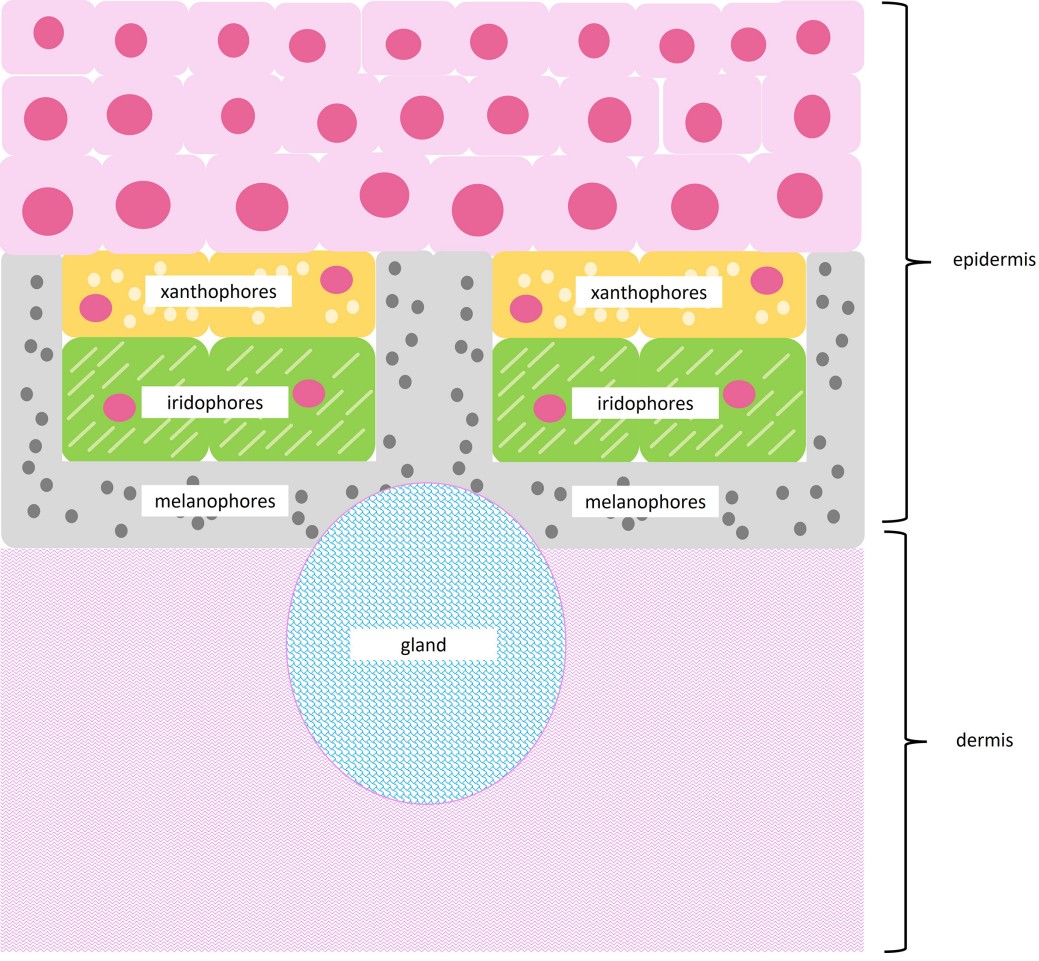

**Figure 2 Subcellular structures of chromatophores.** Illustration of a typical chromatophore unit with subcellular structures. Superficial xanthophores contain pigment vesicles, carotenoid vesicles, and pterinosomes, shown as light yellow circles. Iridophores contain reflective guanine platelets shown as light green dashes. Deep melanophores contain melanosomes, shown as dark gray circles, and may exhibit fingerlike projections that wrap around other chromatophores. The epidermis runs above the chromatophore units, while a thick layer of dermal connective tissue runs below chromatophore units. Glands may be interspersed throughout tissue sections.

chromatophore coverage across the morphs of *R. imitator* may contribute to the understanding of an organism's cellular response to an ecological phenomenon, like aposematism, which exerts divergent selection on a species across its geographic range (*Gallant et al., 2014*).

## MATERIALS AND METHODS

### Slide preparation and image collection

Individuals were collected from populations of each of the four mimetic morphs of *Ranitomeya imitator*. We collected the banded morph near Sauce, San Martin Province; the striped morph near Pongo de Cainarachi, San Martin Province; the spotted morph near Tarapoto, San Martin Province, and the Varadero morph near Varadero, Loreto Province. Individuals were collected during the day by visually locating adults. Six adults

from each color morph (24 in total) were sacrificed within 12 h of collection using 20% benzocaine gel applied to the venter. Immediately after the individuals were sacrificed, their dorsal skins were removed and placed in 10% neutral buffered formalin. All animal handling procedures followed the respective protocols and were approved by Servicio Nacional Forestal y de Fauna Silvestre (SERFOR permiso R.D.G. N° 191-2016-SERFOR-DGGSPFFS and CITES N°17 PE001718) in Peru and the Institutional Animal Care and Use Committee at East Carolina University (AUP D303).

Skins were dissected to separate regions of different colors, either black, yellow, orange, or green. Skin sections of approximately 2 mm × 1 mm were collected from the separated regions. Skin sections were dehydrated, infiltrated, and embedded in paraffin wax using a Tissue TEK VIP 2000 Processor. Reagents and exposure times can be found in Table S1. Finally, a rotary microtome was used to section samples into sections with a thickness of 5 µm before affixing sections to a glass microscope slide (*Amato et al., 2018*). Approximately 1,000 slides were produced, each slide containing 8–10 skin sections.

Histological slides were stained according to Newcomer's Schmorl-Melanin protocol, which was designed to identify sites of melanin deposition. Groups of five slides in a slide basket were dipped in a series of reagents, detailed in Table S1. Critically, the reagents included a potassium ferricyanide solution to stain sites of melanin deposition black and a Nuclear Fast Red solution to make surrounding cells distinguishable. At the end of the series, coverslips were fixed to slides using a Permount® mounting solution.

Histological slides were examined under an Olympus BX-41 Compound Light Microscope with an Olympus DP23 camera attachment and a connection to a computer with Olympus cellSens software. A magnification strength of 40× was used to collect images with a resolution of 72 dpi. The first, fifth, and last skin section from each slide were selected for imaging, so that sections spanned the length of the original skin sample. Slides that had been stained too dark or too light while we were finetuning reagent exposure times from the Newcomer's Schmorl Melanin Protocol were excluded from image collection. Of the images we collected, we chose to exclude from our analysis those images which showed extreme tissue degradation (gaps in tissue exceeded width of epidermis). An example of tissue degradation can be seen in Fig. S1. A total of 2,313 images were used in our analysis. Table S2 provides a summary of the distribution of images across the mimetic morphs, individual frogs, and skin colors.

### Image analysis

Images were analyzed using ImageJ Software. A Bamboo stylus was used to outline regions of different cell types, and the area of each region (in pixels) was calculated. In some cases, several discrete regions had to be added together to find the total area of a given cell type. Per previous research, regions stained with dark brown or black pigmentation below the epidermis but above the dermis were considered to be melanophores (*Bagnara, Taylor & Hadley, 1968*; *Franco-Belussi et al., 2020*; *Frost & Robinson, 1984*). Regions of translucent cellular material with interspersed nuclei between the epidermis and the melanophores were considered to be xanthophores (*Bagnara, Taylor & Hadley, 1968*; *Frost & Robinson, 1984*). The pale region with parallel fibers along the inferior edge of the tissue section was

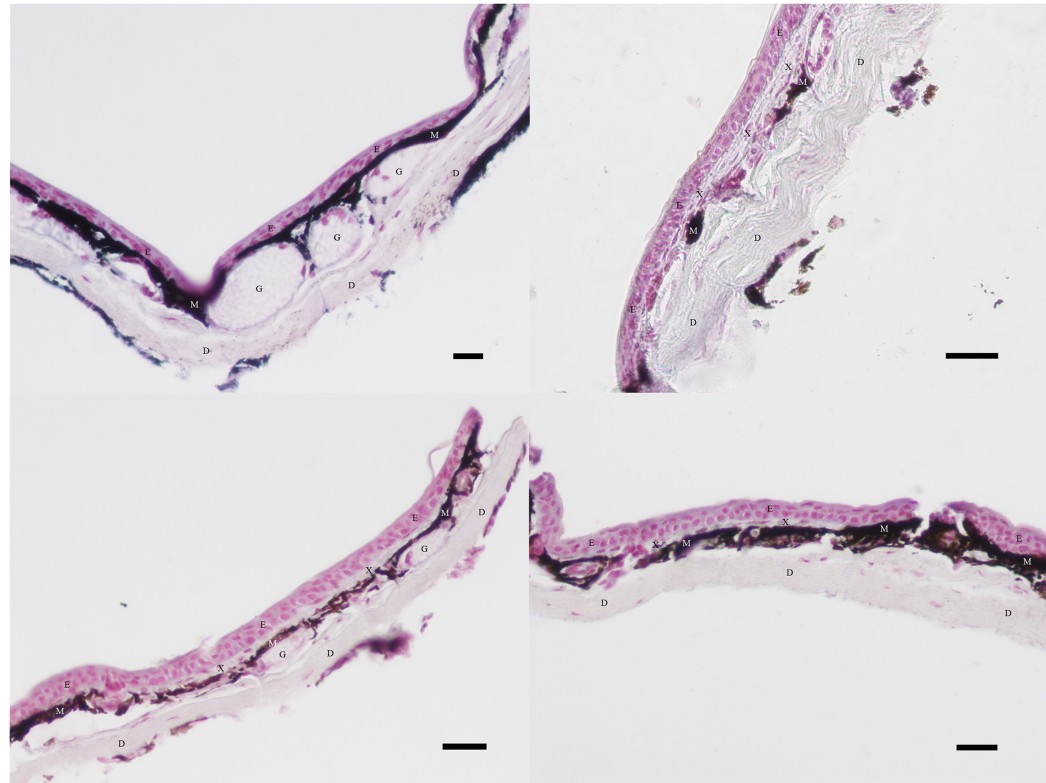

**Figure 3 Examples of excluded and included tissue samples.** Regions of dermis (D), epidermis (E), xanthophores (X), and melanophores (M) visible in skin sections at 40× magnification. Glands (G) are also visible. Clockwise from the upper left, the sections illustrate typical black, orange, green, and yellow skin sections. Xanthophores are present in all skin sections except black. Melanophore coverage varies between colors. Scale bar = 100 μm.

considered to be dermal tissue, and the pink region with densely crowded nuclei along the superior edge of the tissue section was considered to be epidermal tissue (*Bagnara, Taylor & Hadley, 1968*; *Carriel et al., 2011*; *Frost & Robinson, 1984*). Figure 3 provides an example of the cellular structures discernible in sections of black, orange, green, and yellow skin tissue prepared as described in the Slide Preparation section.

For each image, we used ImageJ Version 1.53k to measure the total area of the skin section, the area of melanophores, and the area of xanthophores. Coverage of each chromatophore type was calculated by dividing area of the chromatophore type by total area of the skin section.

Although the data set generated by the present study was large (2,313 images were measured), not all the data points could be considered independent samples (24 adult frogs provided all skin sections imaged). In order to manage the dependent nature of images collected from the same frog, measurements of chromatophore coverage for a given skin color taken from the same individual frog were averaged. Averages were then compared to each other in a one-way ANOVA test in SAS Enterprise Guide 7.1. One-way ANOVA tests were followed up by Tukey's Studentized Range (HSD) test. Whereas the F-values and *P*-values generated by one-way ANOVA tests merely allowed us to support or reject the

null hypothesis that all color morphs belong to the same statistical group, Tukey's Studentized Range test performed pairwise comparisons to quantify the differences in mean between groups and indicated which groups were statistically different and which were not (Tables S3–S5).

## RESULTS

### Black skin

All morphs of *R. imitator* exhibit a dark black dorsal background upon which different colors are presented. Black skin from all morphs was typified by a thick band of melanophores directly below the epidermis and a complete lack of other types of chromatophores. Finger-like projections of melanophores were observed surrounding exocrine glands. The results of the ANOVA suggest no statistically significant differences in melanophore coverage in black skin tissue from the striped, spotted, or banded morphs ($F = 2.190$, $Pr > F = 0.1209$, $df = 3$). Notably, the varadero morph exhibited the lowest melanophore coverage, with approximately 2% less melanophore coverage than the other morphs (Table S3).

### Yellow and orange skin

Of the four morphs of *R. imitator*, three exhibit patches of yellow or orange skin. The striped morph exhibits yellow skin, and the banded and varadero morphs exhibit orange skin (no yellow or orange skin sections were collected from spotted morphs). Yellow and orange skin tissue sections were typified by a thin layer of xanthophores located directly below the epidermis and directly above the melanophore layer. At the level of magnification used in this project, the boundary between the xanthophore and melanophore layers appeared discrete and was not characterized by projections of melanophores into the xanthophore layer. An iridophore layer could not be identified with confidence using our staining procedure.

The results of the ANOVA suggest a statistically significant difference in the coverage of xanthophores among color morphs ($F = 11.7$, $Pr > F = 0.0009$, $df = 3$). The Tukey's HSD test specified a difference between the banded and striped morphs ($\bar{x}_{ba-st} = 5.115$) and the banded and varadero morphs ($\bar{x}_{ba-va} = 4.940$). However, there was no significant difference between the varadero and striped morphs (Table S4). The coverage of melanophores in yellow and orange skin sections was also found to be significantly different across the color morphs ($F = 10.120$, $Pr > F = 0.0017$, $df = 3$), with the striped morph exhibiting the greatest coverage of xanthophores, followed by the varadero morph, then the banded morph (Table S5).

### Green skin

Of the four morphs of *R. imitator*, two exhibit green skin: the striped morph and the spotted morph. Green skin sections were characterized by a thin layer of xanthophores directly below the epidermis and above the melanophore layer. Presumably, iridophores were also present in green skin tissue, but they were not detectable under the staining protocol used in this study.

A one-way ANOVA test indicated a significant difference in the coverage of both xanthophores (F = 21.15, Pr > F = > 0.0001, df = 3) and melanophores (F = 21.11, Pr > F = > 0.0001, df = 3) between green skin sections and the previously examined yellow/orange skin sections. Likewise, the Tukey's HSD test found statistically significant differences in the coverage of xanthophores for all but the striped and varadero morphs (Tables S6 and S7).

## DISCUSSION

Coloration is a key trait in aposematic organisms, although little is known about the mechanisms of color production in polytypic animals. Here we document phenotypic divergence on a cellular level in *R. imitator*, an aposematic and polytypic frog under divergent selection across its geographic range. In documenting histological differences between the four color morphs of *R. imitator*, the results of this study contribute to the growing body of evidence that the color polytypism observed in *R. imitator* and other dendrobatids may be associated with speciation at an early stage (*Gray & McKinnon, 2007*; *Segami Marzal et al., 2017*; *Servedio et al., 2011*; *Twomey, Vestergaard & Summers, 2014*; *Yang, Servedio & Richards-Zawaki, 2019*).

### No variation in melanophore coverage in black tissue

Although some dendrobatids do exhibit dorsal backgrounds of varying darkness (*Posso-Terranova & Andres, 2017*; *Wang & Shaffer, 2008*), no difference in background color has previously been described in *R. imitator*. Moreover, the pressure for *R. imitator* morphs to resemble their models (all of which exhibit dark dorsal backgrounds) would likely act to conserve the dark dorsal background trait within the species. Therefore, it is unsurprising that melanophore coverage in black skin tissue does not vary between three of the four color morphs. Although not a statistically significant difference, the varadero morph was found to exhibit the lowest coverage of melanophores in its black skin tissue, and one previous study found that varadero tadpoles have significantly lower expression of the *mitf* gene, which encodes the melanogenesis associated transcription factor (*Stuckert et al., 2021*). Downregulation of melanogenesis in varadero tadpoles would be consistent with a decrease in melanophore coverage in adult varadero frogs.

### Xanthophores are more abundant and melanophores less abundant in orange tissue than yellow tissue

A previous comparison of spectral reflectance across the four color morphs of *R. imitator* found that the banded morph exhibits the brightest and most contrasting colors (*Twomey et al., 2016*). In the present study, the banded morph was found to contain the greatest coverage of xanthophores and the lowest coverage of melanophores in orange skin tissue, which may be consistent with the production of bright colors. The striped morph exhibited the least coverage of xanthophores, which may indicate that fewer carotenoid/pteridine pigments are required to produce yellow colors than orange colors. Another recent study (*Twomey et al., 2020*) found that the thickness of guanine platelets in the iridophore layer

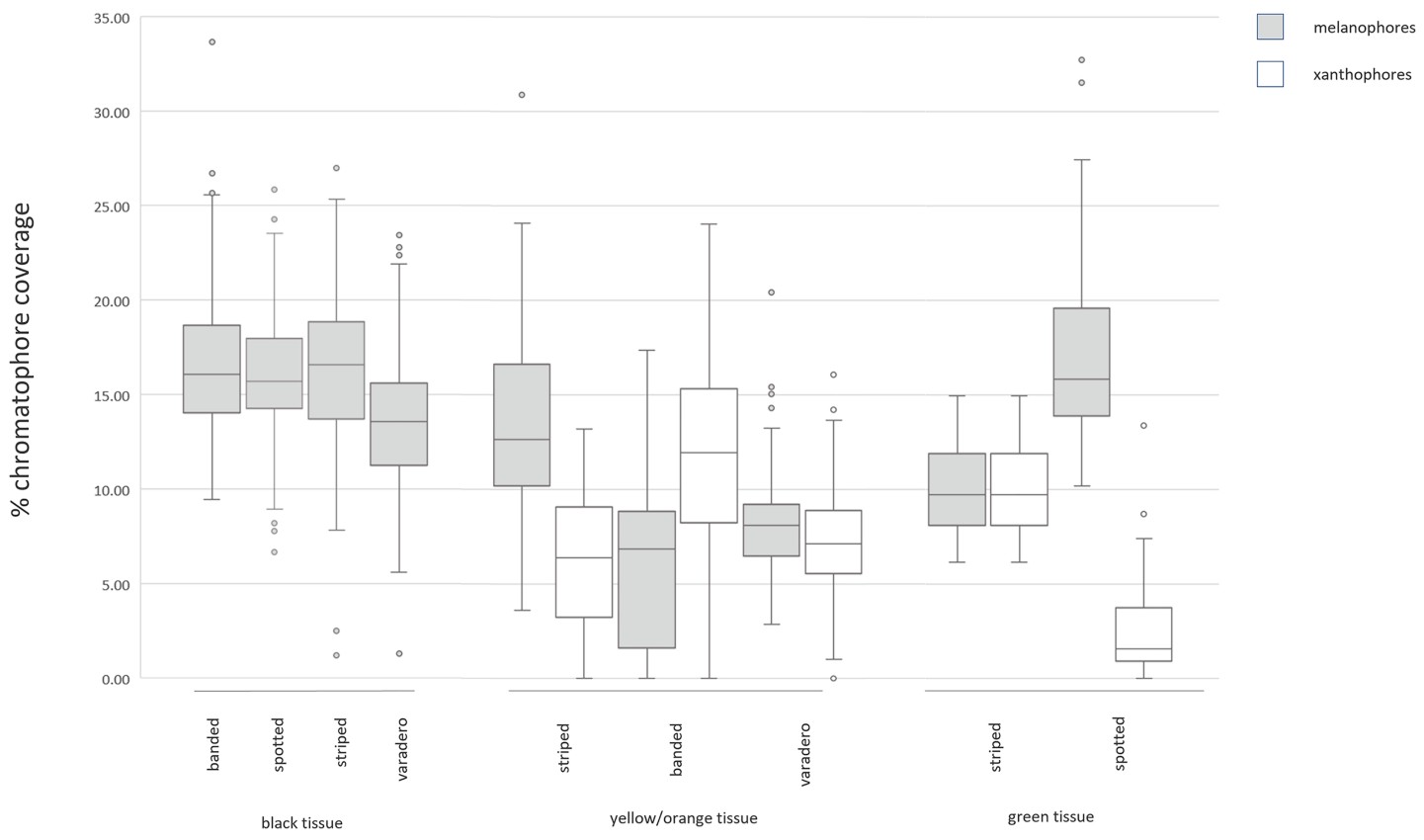

**Figure 4 Labeled examples of specific regions of the epidermis.** Summary of results from measurements of chromatophores across color morphs. Few significant differences exist in black skin tissue across color morphs. In yellow/orange skin tissue, the striped (yellow) morph exhibits lesser xanthophore coverage and greater melanophore coverage than the banded or varadero (orange) morphs. In green skin tissue, the spotted morph exhibits lesser xanthophore coverage and greater melanophore coverage than the striped morph. The morph with brightest colors (banded morph) has the highest ratio of xanthophores to melanophores, and the morph with the dullest colors (spotted morph) has the lowest ratio of xanthophores to melanophores. Tests of statistical significance can be found in Tables S3–S7.

of *R. imitator* skin can also affect hue in the yellow-to-red region of the spectrum, so this may be an additional factor contributing to overall coloration.

## Xanthophores less abundant and melanophores more abundant in green tissue than orange or yellow tissue

Both morphs capable of producing green coloration exhibit a lower coverage of xanthophores than those exhibiting orange colors, which suggests that the production of green coloration may depend less on contributions from carotenoid and pteridine pigments and more on contributions from iridophores and guanine platelets. Frogs capable of producing green coloration also exhibit a higher coverage of melanophores than frogs that produce orange coloration. Previous studies have proposed that melanophore layers may be thickened below iridophores to absorb light that is scattered randomly by the guanine platelets, which may explain the increased melanophore coverage in green skin (*Shawkey & d'Alba, 2017*).

A previous comparison of spectral reflectance across the four color morphs of *R. imitator* found that the spotted morph exhibits the least bright and lowest contrast

colors (*Twomey et al., 2020*). In the present study, the spotted morph was found to contain the lowest coverage of xanthophores and the greatest coverage of melanophores in colored skin tissue, as opposed to the banded morph, which exhibits the brightest and most contrasting colors. Together, these results suggest that a high ratio of xanthophores to melanophores may produce bright colors, whereas a low ratio of xanthophores to melanophores may produce dull colors (Fig. 4).

### Limitations

Two major limitations exist in the present study's methodology: the magnification power used to examine skin sections and the lack of classification of pigments contained in chromatophores. The brightfield microscope used in this study lacked magnification power to identify one of the major chromatophores, the iridophore, with confidence. Likewise, subcellular structures could not be identified with confidence. Previous studies have demonstrated that the size, distribution, and orientation of subcellular structures, like pigment vesicles and guanine platelets, may have significant influence on coloration, but their influence could not be accounted for in the present study (*Frost & Robinson, 1984*; *Posso-Terranova & Andres, 2017*; *Shawkey & Hill, 2005*; *Shawkey & d'Alba, 2017*; *Stuckert et al., 2019*; *Twomey et al., 2020*). Each of the chromatophores described in the present study is capable of producing a variety of pigments. Final skin color may vary depending on the pigment the chromatophore is producing (*Andrade et al., 2019*; *Posso-Terranova & Andres, 2017*; *Twomey et al., 2020*; *Stuart-Fox et al., 2021*). However, the present study did not classify pigments prior to the processing of skin tissue samples for microscopic analysis and thus cannot account for variation in color due to pigment type.

## CONCLUSIONS

*Ranitomeya imitator* provides science with a striking example of color polytypism, produced by the mimic poison frog's need to present aposematic signals familiar to predators across its geographic range. The present study demonstrated that the divergent selection associated with aposematism has led to divergence in the relative coverage of color-producing chromatophores across the four color morphs. Morphs that produce orange skin exhibit a higher coverage of xanthophores and lower coverage of melanophores than those that produce yellow skin. In turn, morphs that produce yellow skin exhibit a higher coverage of xanthophores and lower coverage of melanophores than those that produce green skin. Generally, across the morphs, a high ratio of xanthophores to melanophores can be associated with colors of brighter spectral reflectance.

### Future directions

To date, the majority of studies of coloration in dendrobatid frogs have taken advantage of natural color variations to establish correlations between color production mechanisms and observed color patterns. Studies attempting to experimentally manipulate color production mechanisms would be extremely valuable to the field. For example, the present study found that the spotted morph had a much lower coverage of xanthophores in its green tissue than the banded morph had in its orange tissue. Previous studies have

identified *pax7* and *xdh* as genes associated with the early development of xanthophores and have found both genes to be differentially expressed across the color morphs of *R. imitator* during development (*Stuckert et al., 2021*). If the *pax7* and/or *xdh* gene could be overexpressed in spotted *R. imitator* embryos and the overexpression of those genes led to adult spotted frogs with more abundant xanthophores and a more orange color pattern, there would be direct evidence that increasing xanthophore coverage causes the development of orange skin rather than merely being correlated with it.

## ACKNOWLEDGEMENTS

We thank Andrew Rubio for assistance in maintaining the molecular lab and Eli Bieri for assistance with staining slides. We thank Jeffrey McKinnon and Tim Erickson for advice on this project.

### Funding

This research was funded by the National Science Foundation, Division of Environmental Biology: Evolutionary Processes (Grant# DEB-1655336 to Kyle Summers). There was no additional external funding received for this study. The funders had no role in study design, data collection and analysis, decision to publish, or preparation of the manuscript.

### Grant Disclosures

The following grant information was disclosed by the authors:
National Science Foundation, Division of Environmental Biology: Evolutionary Processes: DEB-1655336.

### Competing Interests

The authors declare that they have no competing interests.

### Author Contributions

- Mallory de Araujo Miles conceived and designed the experiments, performed the experiments, analyzed the data, prepared figures and/or tables, authored or reviewed drafts of the article, and approved the final draft.
- Mikayla Joyce Johnson performed the experiments, authored or reviewed drafts of the article, and approved the final draft.
- Adam M. M. Stuckert conceived and designed the experiments, analyzed the data, authored or reviewed drafts of the article, and approved the final draft.
- Kyle Summers conceived and designed the experiments, authored or reviewed drafts of the article, and approved the final draft.

### Animal Ethics

The following information was supplied relating to ethical approvals (*i.e.*, approving body and any reference numbers):

The East Carolina University Institutional Animal Care and Use Committee provided approval of the protocols used in this research (AUP #D303).

## Field Study Permissions

The following information was supplied relating to field study approvals (*i.e.*, approving body and any reference numbers):

SERFOR (Peruvian Ministry of Natural Resources and Wildlife) and Convention on International Trade in Endangered Species of Wild Fauna and Flora: SERFOR permit R.D.G. N° 191-2016-SERFOR-DGGSPFFS and CITES N°17 PE001718.

## Data Availability

The raw histological data and the aggregate data (averages of the data collected from multiple slides from the same frog) are available at Github and Zenodo: https://github.com/mallorymiles/ranitomeya-histology.

de Araujo Miles, Mallory. (2023). Ranitomeya-histology [Data set]. Zenodo. https://doi.org/10.5281/zenodo.8008204.

## Supplemental Information

Supplemental information for this article can be found online at http://dx.doi.org/10.7717/peerj.15533#supplemental-information.

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
