# Peer review of "A histological analysis of coloration in the Peruvian mimic poison frog (Ranitomeya imitator)"

_PeerJ, doi:10.7717/peerj.15533_

## Round 0.1 · original submission · Major Revisions

The manuscript received two comprehensive reviews. The authors are encouraged to rework the manuscript and resubmit for re-review after they have addressed the comments.

·

Basic reporting

Miles et al. histologically researched the chromatophore composition of four different skin colors and the relative quantity of poison glands across four color morphs of an aposematic dendrobatid frog species. Their main findings are different compositions of the color generating ultrastructures in different skin colors across the four morphs, as well as a higher relative proportion of poison glands in the morph with the highest proportion of aposematic colors.
In general, the research question is interesting and the findings certainly relevant for the field, however I suggest some major revisions particularly on the methodology and the presentation of the results. Additionally, while the chromatophore part is well introduced and discussed, the part on poison glands should be also be revised. What is concerning, and in my understanding not easy explicable by a trivial error, are two contrasting statements on the origin of researched animals. While in the M&M section the authors state they collected animals in the wild and handled them within 24 hours after collection, they delimit their results as not representative for wild populations in the discussion.

Experimental design

The authors have made a substantial effort and worked through a large number of samples. The dataset (6 replicates per morph, totaling to 24 individuals) with more than 2000 individual photos being analyzed, seems suitable to address the research questions. The description of the methodology however is not easy to follow or even incomplete. In its current form, a reader would not be able to reproduce the study. For instance, it remains unclear how many skin samples were analyzed, or how the collection of specimens was conducted (at night, localities, acoustic or visual surveys, size and sex of specimen). The authors should explain the discrepancy with the statement in the discussion (captive animals)!
Information on the used laboratory equipment is mostly lacking. The overall description of the dataset and how it was compiled and standardized is unclear. Potentially, merging the different chapters (e.g. slide preparation and staining, as well as microscopy, imaging and statistical analysis) would help to increase the readability and allow to better understand how the samples where processed and the data generated. I would suggest to add a table summarizing the sampling effort in the M&M section. Particularly, how many skin samples replicates per color and individual have been sampled, the total number of skin samples for each color and the number of analyzed pictures from all individuals and skin colors would be interesting.
For the results, a summarizing table with the actual numbers for the chromatophores and poison glands should be included. The statistics may stay in the supplement; however, authors should give the actual p- values for the results. A graph plotting the poison glands data could also be included.

Validity of the findings

The use of the word ‘abundance’ for the chromatophores and poison glands is somewhat inaccurate and may be misleading. Afterall what has been measured is the relative area that these structures occupy but not their actual numbers. Maybe use ‘coverage’ or ‘proportion’ instead.
While for the chromatophores the coverage is certainly the value of choice to quantify their respective occurrence, I am not convinced this is the best measure for the poison glands. The surface they occupy is a too indirect measure for their quantity, especially as they may vary in size throughout the dorsal skin of an individual, among individuals and potentially even among different morphs. Additionally, we cannot assume that all the glands included in the measurement are actually venom glands / ore are sequestration venom at that stage. I would strongly recommend to divide the gland coverage by the number measured glands. Ideally, a correction to body size should also be included. Although venom glands may not alter in number in post metamorphic life stages, they most certainly hypertrophy throughout seasons (availability of prey), reproductive activity (increased mobility and predation risk). It is unfortunate that the coverage of poison glands in the different skin colors is not compared. Such differences however have been shown in other aposematic, poison sequestrating frogs.
I assume that some of these points can not be addressed without a large amount of additional work. Nevertheless, the up mentioned points and their implication for the results needs to be introduced and discussed in a revised version.

Additional comments

As the manuscript didn’t provide numbered lines, I commented and corrected more elaborately in the in the word file via the comment function and ‘track changes’:
With regard of the above mentioned points, the title should be reconsidered, after all the actual sequestration of toxins is not researched.
Some of the cited literature seem not be ideal choices as references, or not be represented (or not referenced correctly) in the listed literature. Two examples are mentioned in the ‘track changes’ version, there may be more as I have not spent more time on checking them all (given the necessity for some major revisions).
Except for figure 1, none of the figures is referenced in the text.
Figure 1 could get a scale bar (alternatively approximate size (SVL) of a frog should be provided in the test)
Figure 2: Nice illustration! However, not all visible features are explained in the caption. Maybe a bit oversimplified. A more realistic illustration may help to understand the differences better. Added a figure to indicate possible improvements. Alternatively consider showing excerpts from real tissue or combine with figure 4.
Figure 3: Shift to supplement
Figure 4: Maybe provide slides for all 4 different skin colors and indicate differences in chromophore composition with arrows or the like (even if they are very subtle).
Figure 4 should be equipped with a scale bar. It is unclear what skin color the sample represents. D (for Dermis) is not explained in the caption.

·

Basic reporting

I have reviewed the manuscript titled, “A histological analysis of coloration and toxin sequestration in the Peruvian Mimic Poison Frog (Ranitomeya imitator)”. I found it to be a very well written piece of work that was interesting to read and informative. The results expand on our understanding of the interplay between morphology and ecology in poison frogs, particularly in the context of the polytypic R. imitator system. Below I include some concerns and suggestions that the authors might take into consideration to improve the manuscript prior to publication.

One more general concern that arose while reading the manuscript is the issue of the ways in which the authors refer to the chemical ecology vocabulary. There are important differences between the use of the words unpalatable, toxic, noxious, and toxin and in some places it seems as if these were used interchangably when they all refer to distinct chemical and ecological interactions between predators and prey.

Also, while we do know in a general sense that alkaloid toxins are sequestered within the granular glands in the skin in poison frogs, more care could be taken when referring to the specific case of the glands in this study, given that true chemical analysis (GCMS or a MALDI-MSI approach) of the content of the glands was not a part of the study. For example, I question whether the title should refer to toxin sequestration (as a function), or if it would be better to specify that what was studied here were toxin sequestration structures (as a morphological trait) or, more simply, poison (or granular) glands. I would also suggest adding a bit of depth to the information in the section in lines 105-115. There are other arthropods besides ants and mites that also contribute alkaloids by way of the diet that are not mentioned. Also, it seems important to clarify that we do not yet understand very well whether all types of alkaloids are sequestered together within the same granular glands, or if there is separation of different types of alkaloids or the inclusion of other toxins besides alkaloids such as small peptides (volatile organic compounds) in the content of individual granular glands, as occurs in the granular glands of other anurans. The reason to mention this in the introduction would be that a mere histological perspective of the granular glands does not actually inform us about the presence of alkaloids in particular, but rather tells us that there could be a substance in the granular glands that could be comprised of no, some, or all alkaloid-based content. Therefore, it seems that this section of the introduction should offer a more complete background of these perspectives as context for when the reader interprets the results of the study. Papers to consider citing as part of this additional information include Saporito et al. 2007 (https://www.pnas.org/doi/full/10.1073/pnas.0702851104), Mailho-Fontana et al. 2018 (https://frontiersinzoology.biomedcentral.com/articles/10.1186/s12983-018-0294-5), and Gonzalez et al. 2021 (https://link.springer.com/article/10.1186/s12983-021-00420-1).

A major issue that became apparent while reviewing the manuscript is the definition of “poison glands” as any vesicle found between the dermis and epidermis. There are various types of glands present in the skin of amphibians, and the authors do not clarify how they methodologically separated poison glands from mucous glands in this study, which are often present in similar numbers in anuran skin. I recommend reviewing the figures in the paper by Regueira et al. 2016 in The Anatomical Record and/or Angel et al. 2003 in Toxicon about the glands in toad and dendrobatid skin as an excellent references for the distinction between mucous glands and poison glands. There are also many references with relevant images of gland types in papers by Delfino and co-authors in other anurans. If, in fact, the authors have combined poison and mucous glands in the counts of the number of glands in each morph of R. imitator in this study, then it seems to me that the interpretations of these data cannot be applied to toxins and chemical defenses because the proportion of those glands that could hold poison is unknown. Overall, I encourage the authors to consider whether removing the data about glands (or re-analyzing the photos to ensure they really are only counting poison glands) would be a better choice for this manuscript, because the color data on their own are interesting in the context of this system and the recent genetic work in this area.

It also is not clear if the individuals used in the current study are wild-caught (as suggested in the methods, lines 129-132) or captive-reared (as suggested in the abstract and discussion). If they are captive-reared, it might be worth mentioning in the methods or results that there was no expectation that alkaloid/granular content would be found within the glands in histological sections.

Experimental design

- A minor detail, but one that is relevant to the PeerJ reviewing criteria, is that in line 135, the authors mention that the study was approved by both IACUC at ECU and by SERFOR in Perú, but they include only the number of the export permit awarded by CITES and do not list the number of the collection permit awarded by SERFOR, which is a separate institution from CITES.

- Figure 3: It would be useful to use arrows and more specificity in the caption to clarify what exactly is being referenced as “degraded”. However, in a general sense, I’m not convinced that this figure is necessary, given that the decision to use or not use a particular section of tissue is standard practice in histological studies. The quality of the photos in the figure is low, and should be improved if the authors choose to keep the figure. The figure should also indicate 40X if that is the power used for these photos, and it would be helpful to include scale bars.

- It is not clear why 2072 images were taken from 1000 slides with 8-10 skin sections each. It would be helpful to clarify the system that was used to decide which images to include in the analysis of tissue. This is relevant for histological studies because the sampling design can generate significant bias in measurements and counts if sampling is not standardized across animals, tissues, and sections.

- Line 177: More specificity about the types of cells that were outlined should be indicated. Also, it would be useful to give an idea of the size of the measurements in um or mm rather than just in pixels, so that the data can be compared to other studies in the future.

- Line 177-178: How often was it necessary to add regions together, and under which criteria was the decision to add regions together or not made to avoid subjectivity?

- Skin sections can vary greatly in thickness in different regions of the body or dorsum, and the dermis can contain more or less connective tissue at its medial edge in different sections which can affect overall area measurements. How was this taken into consideration in order to ensure that the relative area of chromatophore types and the overall area of a given section were fairly represented across all populations, individuals, and sections in order to allow for statistical comparisons?

Validity of the findings

- Line 187-188: Not all glands are poison glands, as there are also abundant mucous glands present in the same layers of skin but with a somewhat distinct membrane structure. The authors cite “Stynoski 2016” in this line, however that reference does not exist; the images provided in Stynoski and O’Connell 2017 are not enough of a reference in this case because they are images of pre-metamorphic individuals with underdeveloped glands, so a reference that describes glands in adult frogs should be included (such as Regueira et al. 2016 or Saporito et al. 2010). The lack of separation between poison and mucous glands in this study raises serious concerns that the data provided in lines 266-270 are not reliable, and accordingly that the findings are not relevant to the differences in chemical defenses among color morphs.

- The heading in line 334-335 suggests to the reader that in this study the sequestered toxins were quantified, although as the study was described, the toxins were not measured, only glands.

- From the evidence presented, the conclusion that the varadero morph can sequester more toxins seems inappropriate and/or premature. The finding that the varadero morph has 10.4 rather than 4.5 glands per section of skin is interesting, but could be a reflection of (1) different numbers of mucous or glandular glands and thus not necessarily a reflection of toxin sequestration, (2) differences in the amount of chemical defenses stored in glands made up of compounds other than alkaloids, or (3) differences in the size of individuals in populations or length of tissue sections in a given morph (which were not corrected for in the measure of abundance, as least as far as was described in the methods). Additionally, the number of glands is not necessarily the best measure of the amount of substances that are present in the glands, because gland size tends to vary enormously and some granular glands become large and distended when full of granular content (presumably alkaloids), while others remain small, even within the same tissue sections. See Saporito et al. 2010 (and its figures 4, 5 and 6) for a discussion of this size versus quantity issue as well. This is especially true given that the frogs in this study were captive-reared and thus unlikely to have glands filling up with chemical defenses, as mentioned in the discussion.

Additional comments

- It appears that there is an error in the listed affiliation of the first author in the PeerJ system (“East African University”).

- Line 55: Remove “level” after population (or change “within a” to “at the”).

- Line 69: Change “share” to “shares”.

- Line 111: There is no reference in the References for Stynoski 2016. Most likely, “Stynoski 2016” should be changed to “Stynoski and O’Connell 2017” in the text (also in lines 112 and 188).

- Line 114: It might be better to clarify that, rather than metabolic alteration being “rare”, that it is only known for a very small number of the hundreds of types of alkaloids that are sequestered by the frogs. The chemical alteration could be common, but only applied to a few of the many chemical compounds that are sequestered.

- Line 516: Change “Ecological Letters” to “Ecology Letters”.

- Line 129: Change to “collected from populations of each mimetic morph”.

- Line 132: Specify in the references or in the text what the “AUP D303” refers to here?

- In line 133, the authors mention that skins were placed in “neutral buffered formalin” and then in line 141 they mention that the skins were separated and then fixed in “phosphate buffered formalin”. It doesn’t seem like the authors would fix the tissue twice, so I’m guessing that some text needs to be removed from line 133?

- More specificity is needed in line 142 regarding the tissue processing protocol, or at the least a reference is needed to cite the protocol that was used. It could be that this information is referenced in the supplementary info (Table 7?) but if that’s the case, the table needs to be referred to in the text in place of the very general description of “several dehydrating and clearing solvents”.

- In line 142-143, the rotary microtome would only have been used for sectioning tissue, and not for embedding and attaching to slides, correct?

- Line 160: Information about the brand of microscope and camera are needed here.

- Line 164: The word “inviable” refers to “unable to survive” and “viable” means “able to survive”. Suggested to choose other terms for this sentence that refer to the use or viewing of the tissue section.

- Line 194: Remove “using”.

- Line 195: Not necessary to mention that data were recorded in Excel.

- Lines 205-215: Not necessary to explain in such detail the objectives of ANOVA and Tukey tests in a general sense, but rather just mention that they were used, because these are very standard tests.

---

## Round 0.2 · Minor Revisions

Dear Authors
Your manuscript has been re-reviewed, and the reviewer made several small recommendations. Please address these.

I suggest you also try to address the comment re "actual implications for the species and its outstanding mimicry..." as this may increase the citations and discussion value of the manuscript.

Kind regards

Annemarie Avenant-Oldewage

·

Basic reporting

In the revised manuscript “A histological analysis of coloration in the Peruvian Mimic Poison Frog (Ranitomeya imitator)” by Miles et al., the authors compared the color generating ultrastructure in differently colored skin sections across the four color morphs of an aposematic dendrobatid frog species. The authors show that the two most important color producing structures, melanophores and xanthophores, vary in abundance/coverage across the different color morphs.
The authors addressed all major critical points of the previous version. The decision to exclude the ‘poison’ gland part version is understandable and, from a scientific point of view, probably the right choice.

The major criticism I would state is that the discussion does not really tie back to the study objective(s) stated in the introduction. It is mainly discussed how the observed pattern may come about, but not what the actual implications for the species and its outstanding mimicry are (i.e. different morphs may have varying metabolic costs in order to maintain their respective aposematic coloration). However, this may be a question of "taste" and does reduce the relevance of the findings.

After addressing the minor corrections specified below, and with regard to the journals aims, the manuscript should be considered for publication.

Experimental design

The extensive dataset is suitable to address the research questions. The methodology is (now) well explained, easy to follow and presented in a traceable manner. The manuscript certainly fills a knowledge gap and provides a basis for relevant follow up questions.

Validity of the findings

The statistical approach suits the dataset with only small exceptions (specified in the additional comments). These exceptions however, do not have any consequences for the interpretation of the results. I recommended displaying the actual p-values for the respective tests in the tables in the previous version of the manuscript, a point probably overlooked by the authors. I still would still be good to display the actual p-values in the respective tables.

Additional comments

Line 135: add trademark disclosure to Permount
Line 141: too dark or too light? - I am not a native speaker but “too lightly and darkly” sounds somewhat unusual to me
Line 157: specify what pale region (e.g.: “In the slides, pale regions with…”) or ad reference to Figure 3
Line 194 -195: Tukey test =Post Hoc test! Preconditions for the test are not met when the ANOVA indicate no difference! Alternatively, do a pairwise comparison with a non-parametric method (e.g. wilcoxon test) but then you need to manually adjust the p-value. Questionable whether this would give you a significant result... better to not state the statistical significance of that difference!
Line 191: delete ‘poison’ or be more general, like "exocrine skin glands"
Line 208: Pr > F
Line 209: specified
Line 281: see comment for line 194-195

---

## Round 0.3 · Minor Revisions

Please find the comments of Reviewer 2 on the 2nd draft below. Will you please address these and list your actions in table format along with the requests? The manuscript may be returned to this reviewer for final approval.
Kind regards

Annemarie Oldewage

·

Basic reporting

I’ve now read through the revised version of the manuscript titled, “A histological analysis of coloration in the Peruvian Mimic Poison Frog (Ranitomeya imitator)”. It appears that the version of the rebuttal letter and track changes that I can download in this revision did not include indications of the points in the comments that I made on the first version (they only indicated the responses to the other reviewer?). But, I can see that numerous other changes were made to the manuscript that coincide with my comments to the first submission. Overall, it is very well written and easy to read.

Primarily, the authors have chosen to remove the analysis of gland data and throughout the manuscript have reduced the focus on toxins and poison glands and instead emphasized the information about color. Due to the issue of distinguishing poison vs. mucous glands in the first version of the manuscript, I think this was a good choice, and in some ways even streamlined the focus of the paper to highlight the mechanisms behind color evolution via histological data in R. imitator.

Now that the manuscript is entirely focused on integumentary mechanisms of color production among morphs, it seems like it would be helpful to have a short paragraph in the introduction that explains how much is known (or not known) about the histological perspective of color production in dendrobatids (and/or other anurans). Lots of the work that has been done on dendrobatid color production is genomic/genetic. However, there is microscopy work in this species (Twomey et al. 2020) and other members of this or related frog families that would be helpful to the reader to mention the relevant work and have this context in the introduction so as to be able to contextualize/judge the novelty of the presented findings (to directly clarify the gap in knowledge that this study addresses...what is new about this information relative to other studies?). Some of this context is addressed in different parts of the discussion, but it would also be helpful to “set the stage” with the dendrobatid literature in the introduction to justify the study’s objectives.

It may also help to finish the introduction with the statement of a hypothesis or prediction about chromatophore coverage in the different skin colors and morphs, so that the expectations of the study are clear from the start. Would it be expected that melanophore or xanthophore coverage would differ among the four morphs, for example? If so, why or why not?

Experimental design

Methods were easier to follow and more detailed in this version, thanks for clearing up many large and small points.

Validity of the findings

Just one minor thing: In line 244, the authors state that “the Varadero morph was found to exhibit the smallest area of black skin on its dorsum”. However, the area of black skin on each frog (more black patterning on an organismal level) was not the variable of interest in this study, but rather the amount of melanophore coverage (black-chromatophore density within already black tissue sections on a cellular level), correct? This statement in the discussion should be adjusted so that it reflects the particular variable that was measured in the study.

Additional comments

In my version in the reviewing platform, there is still a supplemental table (6b) that has the title “Tests for abundance of poison glands in skin” that I imagine the authors will want to change before publishing to avoid confusion.

Line 104 – It doesn’t seem necessary to mention the statistical methods at the end of the introduction, especially because they are standard statistical tests. Those details could be left for the methods section.

---

## Round 0.4 · accepted · Accept

I assessed the response to the reviewers and am satisfied that the comments have been addressed satisfactorily.